# Cas9-AAV6 gene correction of beta-globin in autologous HSCs improves sickle cell disease erythropoiesis in mice

Adam C. Wilkinson [1,2,5✉], Daniel P. Dever [1,3,5], Ron Baik[1,3], Joab Camarena [1,3], Ian Hsu[1,2], Carsten T. Charlesworth[1,2], Chika Morita [1,2], Hiromitsu Nakauchi [1,2,4✉] & Matthew H. Porteus [1,3✉]

CRISPR/Cas9-mediated beta-globin (*HBB*) gene correction of sickle cell disease (SCD) patient-derived hematopoietic stem cells (HSCs) in combination with autologous transplantation represents a recent paradigm in gene therapy. Although several Cas9-based *HBB*-correction approaches have been proposed, functional correction of in vivo erythropoiesis has not been investigated previously. Here, we use a humanized globin-cluster SCD mouse model to study Cas9-AAV6-mediated *HBB*-correction in functional HSCs within the context of autologous transplantation. We discover that long-term multipotent HSCs can be gene corrected ex vivo and stable hemoglobin-A production can be achieved in vivo from *HBB*-corrected HSCs following autologous transplantation. We observe a direct correlation between increased *HBB*-corrected myeloid chimerism and normalized in vivo red blood cell (RBC) features, but even low levels of chimerism resulted in robust hemoglobin-A levels. Moreover, this study offers a platform for gene editing of mouse HSCs for both basic and translational research.

[1] Institute for Stem Cell Biology and Regenerative Medicine, Stanford University School of Medicine, Lorry I. Lokey Stem Cell Research Building, 265 Campus Drive, Stanford, CA, USA. [2] Department of Genetics, Stanford University School of Medicine, Stanford, CA, USA. [3] Department of Pediatrics, Stanford University School of Medicine, Stanford, CA, USA. [4] Division of Stem Cell Therapy, Distinguished Professor Unit, The Institute of Medical Science, The University of Tokyo, Tokyo 108-8639, Japan. [5] These authors contributed equally: Adam C. Wilkinson, Daniel P. Dever. ✉email: adamcw@stanford.edu; nakauchi@stanford.edu; mporteus@stanford.edu

Allogeneic hematopoietic stem cell transplantation (allo-HSCT) is a curative therapy for a wide range of hematological diseases including sickle cell disease (SCD)[1,2]. Unfortunately, even with the identification of immunologically well-matched donors (which most patients do not have), HSCT recipients can still suffer from graft-vs-host disease and/or the consequences of long-term immunosuppression. However, allo-HSCT using hematopoietic stem cells (HSCs) from sickle cell trait donors has demonstrated that even a single *HbA* allele (the common healthy variant) can cure the disease. The advent of CRISPR/Cas9 genome editing has opened the potential for efficient correction of disease-causing genetic mutations in patient-derived HSCs, including the A-to-T base mutation within codon 6 of the *HBB* gene in SCD patients that is responsible for the mutant *HbS* allele and the resulting glutamic acid to valine (E6V) amino acid substitution within beta-globin[3–5]. The combination of gene-correcting patient HSCs and autologous HSCT offers a potentially revolutionary approach to cure SCD and a range of other hematological diseases.

A number of Cas9-based gene correction approaches for *HBB* and other genes have now been validated in human hematopoietic stem and progenitor cells (HSPCs) and are rapidly progressing toward clinical trials[3,6,7]. One of the most promising technologies for gene-correcting human HSPCs involves Cas9-mediated genome cutting in combination with recombinant adeno-associated virus serotype 6 (AAV6) homologous recombination (HR)-based repair[3,8–11]. However, to date the repopulation function of gene-corrected human HSPCs modified with Cas9 has only been assessed using xenograft transplantation into immunodeficient mice and no studies have evaluated these disease-correcting reagents in the context of autologous transplantation for SCD. While xenograft assays are useful models to study human HSPCs in vivo, the xenogeneic environment limits their utility. In particular, xenograft assays poorly support the development of mature human red blood cells (RBCs), which restricts the assessment of gene correction of erythropoietic defects to in vitro studies.

Over 20 years ago, Townes and colleagues generated a transgenic mouse model of SCD by humanizing the hemoglobin locus to carry both human *HBA* and the SCD mutant (E6V) form of *HBB* (*HbS*)[12]. These *HbS* Townes-SCD mice only express human sickle hemoglobin tetramers (HgbS) and exhibit many of the symptoms of SCD, including short RBC half-life, RBC sickling, high peripheral blood (PB) reticulocyte frequencies, and radiation sensitivity. Notably, these mice have advantages over other SCD mouse models for gene correction studies, such as the Berkeley-SCD model[13], because Townes-SCD mice were generated by targeted gene replacement rather than random multicopy integration of human globin genes in the mouse globin-null background. The Townes-SCD mouse model therefore allows for the use of beta-globin gene editing reagents optimized for use on human genetic sequences. In addition, it has been previously shown that HR-based repair of *HbS* to *HbA* within mouse embryonic stem cells from Townes-SCD mice normalizes erythropoiesis in re-derived mice[14]. However, although provided to the community as a tool to evaluate SCD therapies, to date only one autologous transplantation approach has been published using the Townes-SCD model[15].

Here, we evaluate HR-based correction of the human beta-globin gene via Cas9-AAV6 technology in Townes-SCD mouse HSCs. Cas9-AAV6-mediated beta-globin correction led to long-term stable hemoglobin A (HgbA), with a corresponding decrease in HgbS, following autologous transplantation into Townes-SCD recipient mice. Engraftment of gene-corrected HSCs reduced RBC sickling and reticulocyte frequencies in the PB while increasing in RBC half-life. This study therefore offers a platform for gene editing of mouse HSCs for both basic and translational research.

## Results

**Cas9-AAV6 gene targeting in functional LT-HSCs.** We recently developed a culture media that supports mouse functional HSCs ex vivo[16,17]. To initially investigate whether this system could also support HR-based gene editing of mouse HSCs by Cas9-AAV6 technology, we tested HR at the *Rosa26* safe harbor locus (Fig. 1A) using a previously validated sgRNA[18,19] pre-complexed with HiFi Cas9 protein[11] and an AAV6-repair template containing a GFP expression cassette. We initially compared two editing strategies: (1) gene editing of bulk mouse bone marrow (BM)-derived cKit-enriched HSPCs after 48 h culture and analysis on day 4 (similar to current human CD34+ HSPC gene editing protocols[3,11]) and (2) gene editing of 7-day mouse HSPC cultures initiated from CD150+Kit+Sca1+Lin− BM HSCs with a media change on day 6 and analysis on day 14. The day-7 HSPC gene-editing protocol achieved higher GFP^hi frequencies, averaging 25% using this strategy (Fig. 1B, C). By contrast, gene targeting of day-2 cKit-enriched HSPC cultures only achieved an average of ~10% GFP^hi cells (Fig. 1C).

Given the benefit of the 7-day pre-culture of mouse HSCs, we focused on this protocol in our next experiments. We directly compared the flow cytometric frequency of GFP^hi cells and droplet digital PCR (ddPCR) quantification of genomic-level *Rosa26*-targeted GFP cassette alleles. *Rosa26-GFP* alleles were on average 1.3-fold higher than the frequency of GFP^hi cells analyzed, suggesting that the majority of targeted cells were monoallelic (Supplementary Fig. 1A). As expected, GFP alleles and the GFP^hi population were not detected in the controls (mock, RNP-only, AAV-only) (Supplementary Fig. 1A). To evaluate whether functional HSCs had been gene targeted at the *Rosa26* locus, we performed transplantation assays and tracked donor cell engraftment in recipients (Fig. 1A). High donor CD45.1 chimerism was observed in the recipients (Supplementary Fig. 1B) and we could detect a GFP^hi cell population in all PB cell lineages (Fig. 1D, E). Importantly, this not only included neutrophil/monocytes, T cells, and B cells, but also RBCs and platelets (PLTs). Secondary transplantation studies further confirmed gene targeting of long-term (LT)-HSCs could be achieved using Cas9-AAV6 technology, with GFP^hi cell donor chimerism observed in all PB lineages in secondary recipients (Fig. 1F). In addition, ddPCR molecular analysis of *Rosa26* alleles confirmed GFP knockin in these secondary recipients (Supplementary Fig. 1C).

In a replicate experiment, we again observed high donor chimerism in recipients (Supplementary Fig. 1D), but greater heterogeneity in GFP^hi cell chimerism between primary recipients (Supplementary Fig. 1E), although the average percentage engraftment of gene-targeted alleles was again similar to the pre-transplanted HSPC population. However, we could still confirm engraftment of edited HSCs in secondary recipient mice (Supplementary Fig. 1F, G). The heterogeneity in the primary recipients suggests that gene-edited functional HSCs may be at low abundance, similar to the recently observed oligo-clonality of human gene-edited HSCs[20,21]. Consistent with the idea that gene editing caused some toxicity within HSCs, gene editing reduced total donor engraftment levels in competitive transplantation assays (against $1 \times 10^6$ whole bone marrow cells) by >30% long-term (Supplementary Fig. 1H), even though few differences between mock and RNP+AAV-edited HSCs could be observed in the ex vivo cell cultures (Supplementary Fig. 1I–K). An important finding from this autologous syngeneic work is that, in contrast to the human into mouse xenogeneic experiments, there was not a

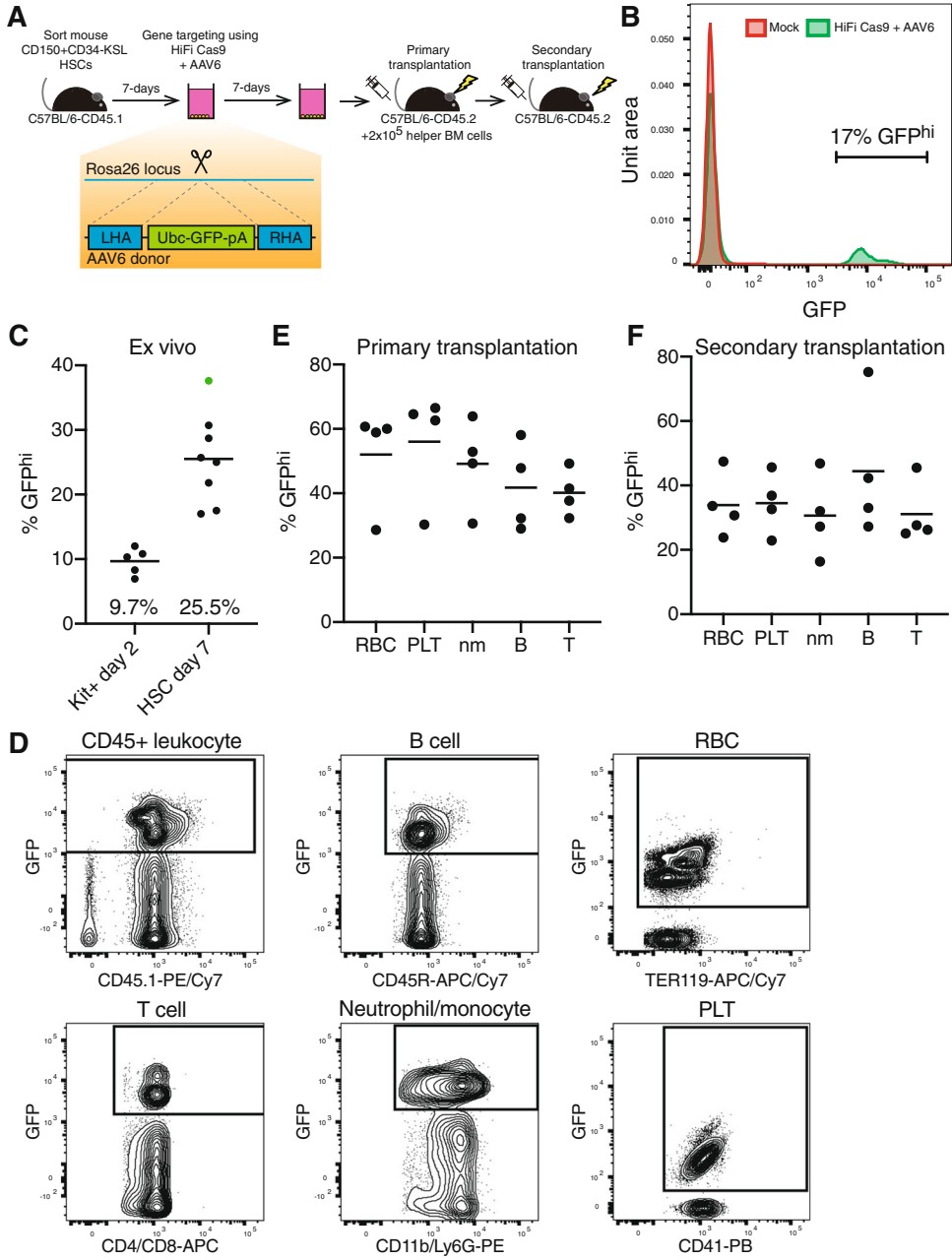

**Fig. 1 Cas9-AAV6 gene editing of functional multipotent LT-HSCs. A** Experimental schematic for introducing a GFP expression cassette into the *Rosa26* locus of mouse hematopoietic stem and progenitor cells (HSPCs) via Cas9-AAV6 gene targeting and functional assessment by transplantation assays. **B** Representative flow cytometry histogram of GFP expression from day-14 HSPC cultures. **C** Mean frequency of GFP[hi] cells from *Rosa26* gene editing schematized in (**A**) when performed on day-2 cKit-enriched bone marrow (BM) cells or day-7 HSC-derived cultures, measured by flow cytometry ($n = 5$ and $n = 8$ independent cell cultures, respectively). HSPC culture highlighted in green used for transplant studies described in (**E**). **D** Representative flow cytometric plots for GFP expression within peripheral blood (PB) CD45[+] leukocyte, CD45R[+] B cell, and CD4/CD8[+] T cell, CD11b/Ly-6G[+] myeloid neutrophils/monocyte (nm), Ter119[+] red blood cell (RBC) and CD41[+] platelet (PLT) populations in transplant recipients described in (**A**). **E** Mean frequency of GFP[hi] cells in PB lineages of primary recipients at 20-weeks post-transplantation, measured by flow cytometry. Day-14 cultures (derived from 200 HSCs; ~2 × 10⁵ cultured cells) transplanted alongside 2 × 10⁵ helper whole bone marrow cells (WBMCs) per recipient ($n = 5$ mice). **F** Mean frequency of GFP[hi] cells in PB lineages of secondary recipients at 12-weeks post-transplantation, measured by flow cytometry. 1 × 10⁶ WBMCs collected from all primary recipient mice (**E**) were transplanted per secondary recipient ($n = 4$ mice). Source data are available in the Source data file.

significant decrease in gene-targeted alleles following primary and secondary transplantation.

**HBB repair correlates with PB HgbA and reticulocyte counts.** Having confirmed that we could perform precise gene targeting in functional mouse HSCs, we next evaluated the functional

consequences of *HbS* gene correction in the Townes-SCD model[12]. Although generated over 20 years ago, few papers have used this model to perform HSCT into SCD mice[15], but instead often transplant SCD mouse-derived HSCs into wild-type (e.g., C57BL/6) mice[22,23]. Since autologous HSCT is key to evaluating how gene-corrected HSCs engraft and function within the diseased recipient, we first needed to establish an autologous HSCT protocol using the

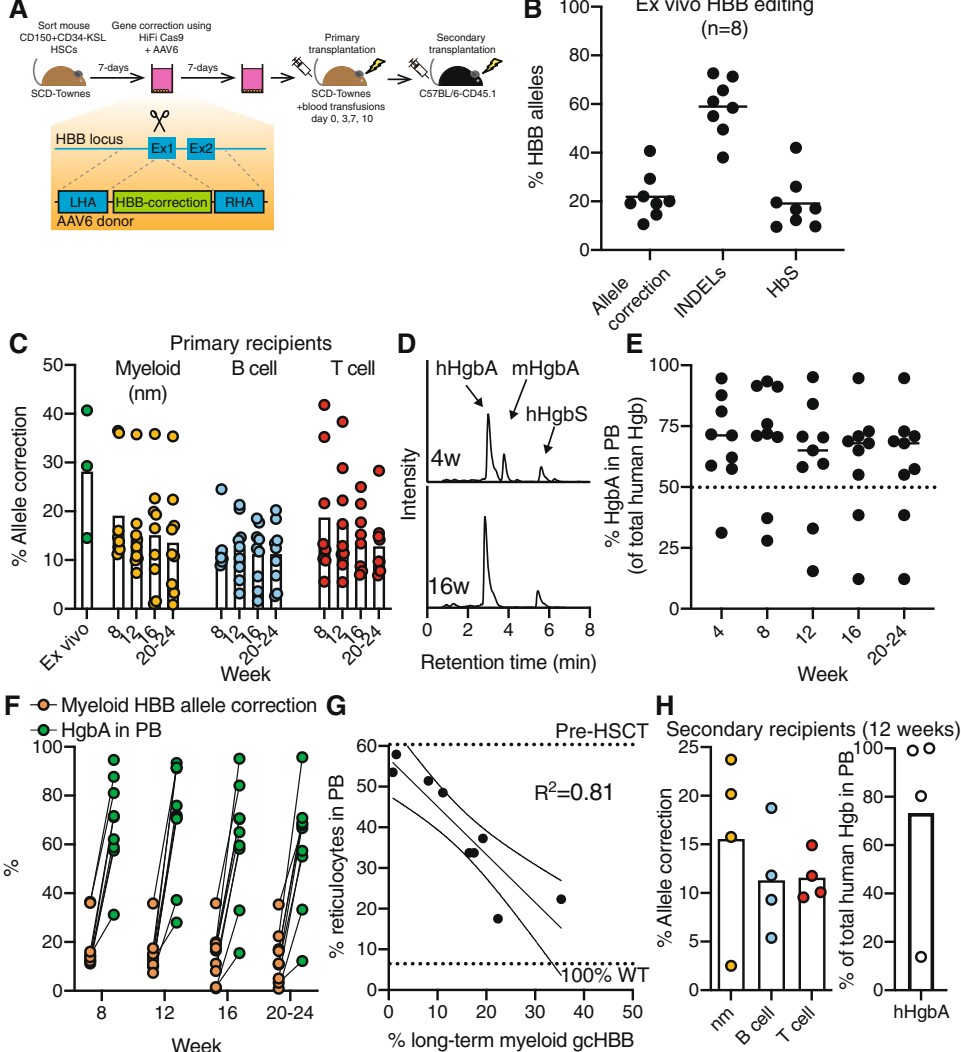

**Fig. 2 Stable HgbA production in SCD mice autologously transplanted with E6V-corrected HSPCs. A** Experimental schematic for Cas9-AAV6-based gene correction of the E6V mutation in beta-globin in Townes-SCD mouse HSPCs. **B** Mean *HBB* gene editing outcomes in mouse HSPC cultures from 8 biological replicates. **C** Mean *HBB* allelic correction frequencies in PB myeloid (nm), B cell, and T cell lineages over time in primary recipients ($n = 9$ mice). **D** Representative HPLC plots for hemoglobin tetramer analysis from PB collected at 4- and 16-weeks post-transplantation, with mouse HgbA (mHgbA), human HgbA (hHgbA), and human HgbS (hHgbS) annotated. **E** Mean frequency of hHgbA (of total hHgbA and hHgbS) in HSCT recipients ($n = 9$ mice). Dotted line represents 50% hHgbA. **F** Mean frequency of PB myeloid (nm) *HBB* allelic correction and hHgbA over time in HSCT recipients ($n = 9$ mice). **G** X–Y linear correlations between PB reticulocyte frequency and myeloid (nm) gene correction at 16-weeks post-transplantation ($n = 9$ mice). **H** Mean frequency of *HBB* allelic correction in PB leukocyte populations (left plot) and frequency of human HgbA in the PB as a total of human Hgb (right plot) in secondary recipients at 12-weeks post-transplantation ($n = 4$ mice). $1 \times 10^6$ WBMCs from mouse A or mouse B (see Table 1) were transplanted into recipient mice (2 mice per primary recipient). Source data are available in the Source data file.

Townes-SCD mouse model. We found that 9.5 Gy radiation was required to clear endogenous erythropoiesis within 4 weeks based on using the amount of sickle hemoglobin (hHgbS) present in the PB as a surrogate for niche clearance with a dose-response of engraftment proportional to the amount of irradiation (with lower radiation doses leading to lower levels of hHgbA) (Supplementary Fig. 2A). However, consistent with the reported short (>3-day) half-life of RBCs in Townes-SCD mice[24], 9.5 Gy radiation caused a significant increase in mortality within a week. This was not noted in the only previously reported autologous transplantation assay, performed by Hanna et al., but this discrepancy may be due to strain differences, irradiator calibration differences, or even more likely differing donor cell type (this previous report used pluripotent stem cell-derived HSPCs)[15]. To perform autologous HSCT using gene-corrected adult bone marrow HSCs, we established an HSCT plus mouse blood transfusion experimental strategy, where 9.5 Gy

irradiated and transplanted Townes-SCD recipient mice were supported by four blood transfusions between day 0 and 10 to exogenously support erythropoiesis during the lag between donor HSPC engraftment and eventual production of the endogenous RBC compartment (Fig. 2A). We initially validated this model by transplanting HSPCs from *HbA* mice[14] into *HbS* carrying Townes-SCD mice. Stable HgbA was observed in the peripheral blood (Supplementary Fig. 2B) and hematological parameters improved in these mice (Table 1), in particular PB reticulocyte frequencies, which dropped from an average of ~60 to ~9%.

Using HiFi Cas9, sgRNA, and AAV6 reagents previously developed and validated for human *HBB*[3,11,25], we next evaluated *HBB* gene correction of the A-to-T mutation in exon 1 of *HbS* (causing the E6V amino acid change responsible for SCD) in Townes-SCD mouse HSPCs. Using an established ddPCR assay[11,26], we detected allelic correction efficiencies at an average

### Table 1 Hematological parameters in Townes-SCD mice.

| | % hHgbA | RBC (×10$^6$/μl) | HGB (g/dl) | % Retics | PLT (×10$^3$/μl) | WBC (×10$^3$/μl) | % myeloid *HBB* HR correction | % myeloid *HBB* INDELS |
|---|---|---|---|---|---|---|---|---|
| Pre-HSCT (*n* = 7) | 0 | 7.0 ± 0.8 | 11.5 ± 1.1 | 60.7 ± 4.6 | 756 ± 159 | 61.1 ± 16.4 | N/A | N/A |
| *HbA* into *HbS* (*n* = 4) | 100 | 9.4 ± 1.1 | 11.6 ± 0.6 | 8.6 ± 0.4 | 949 ± 230 | 13.6 ± 8.9 | N/A (100% *HbA*) | N/A (100% *HbA*) |
| *HbS* into *HbS* (*n* = 2) | 0 | 6.4 | 10 | 51.5 | 1264 | 58.7 | N/A (100% *HbS*) | N/A (100% *HbS*) |
| Mouse A | 84.1 | 7.18 | 8.9 | 22.3 | 1380 | 18.3 | 35.4 | 9.6 |
| Mouse B | 95.1 | 6.55 | 8.3 | 17.5 | 1391 | 13.9 | 22.4 | 21.0 |
| Mouse C | 33.0 | 4.05 | 6.9 | 53.5 | 931 | 53 | 1.0 | 82.6 |
| Mouse D | 58.2 | 4.39 | 6.9 | 48.5 | 716 | 50.8 | 11.1 | 79.5 |
| Mouse E | 59.6 | 3.23 | 5.6 | 51.4 | 737 | 51.7 | 8.2 | 81.2 |
| Mouse F | 70.4 | 4.96 | 7.2 | 37.3 | 851 | 33.8 | 19.3 | 68.3 |
| Mouse G | 70.7 | 4.74 | 7 | 33.7 | 881 | 21.3 | 16.5 | 71.2 |
| Mouse H | 15.5 | 3.37 | 5.7 | 57.9 | 473 | 61.2 | 1.5 | 79.5 |
| Mouse I | 65.0 | 3.83 | 5.6 | 33.7 | 1307 | 44.2 | 17.5 | 63.0 |

Hematological parameters for pre-transplanted Townes-SCD *HbS* mice, 16-week post-transplantation for *HbS* recipients, and 16-week post-transplantation for individual recipients of *HBB* gene-corrected Townes-SCD HSCs. In all cases, recipient mice were *HbS* Townes-SCD mice. Where displayed, averages are mean ± standard deviation. Retics %, percentage of PB reticulocytes.

of ~20%. A similar proportion of alleles (~20%) remained unedited *HbS*, while the INDELs were detected in the remaining ~60% (Fig. 2B). From ddPCR of colonies generated in methylcellulose assays after editing, we estimate ~20% were monoallelic correction (all HR/INDEL, and no WT/HR detected) while another ~6% were biallelic correction (HR/HR) (Supplementary Fig. 2C). Gene correction was also confirmed by Sanger sequencing (Supplementary Fig. 2D). Although the allelic *HBB* correction was lower than the efficiencies seen in human HSPCs[3,11], we hypothesized that this might still be sufficient to correct erythropoiesis in SCD mice. It has been reported previously that in the allo-HSCT setting, as low as 20% myeloid chimerism can be curative[27,28]. To evaluate the repopulation capacity of *HBB*-corrected functional HSCs, we performed ex vivo gene correction followed by radiation-based conditioning and autologous HSCT (Fig. 2A).

Transplanted mice were followed for every 4 weeks post-transplantation to evaluate the retention of gene-targeted alleles in multiple differentiated hematopoietic lineages, hemoglobin expression (both human and transfused mouse), and reticulocyte generation. To quantify chimerism of *HBB*-corrected HSCs in this autologous setting, we performed ddPCR on FACS-purified PB populations (CD11b/Ly-6G$^+$ myeloid neutrophil/monocyte cells, CD4/CD8$^+$ T cells, and B220$^+$ B cells) from 8-weeks onward (Fig. 2C). Although we observed differences between individual recipients, multilineage allele correction could be detected in all recipients long-term (Fig. 2C). This heterogeneity in chimerism was similar to our *Rosa26* transplantation experiments. However, unlike the *Rosa26* transplantation experiments, small reductions in chimerism were seen in vivo as compared with the input sample. INDELs could also be stably detected within these lineages (Supplementary Fig. 2E).

We quantified hemoglobin tetramers in the PB over time by HPLC to determine whether *HBB* gene correction resulted in stable human adult hemoglobin (hHgbA) production. HPLC could distinguish not only corrected-hHgbA from mutant-hHgbS, but also mouse hemoglobin A (mHgbA) introduced by the blood transfusions (Fig. 2D). While mHgbA could be detected at 4-weeks post-transplantation (18-days after the final blood transfusion), only human hemoglobin tetramers could be detected from 8-weeks onward. In most HSCT recipients (7/9 mice transplanted with gene-corrected HSCs), the dominance of HgbA was evident and was sustained above or around 50% of the total human hemoglobins (hHgbA + hHgbS) for at least 20 weeks (Fig. 2E). In this pan-cellular situation, >50% hHgbA is important because it is close to what is found in the healthy sickle cell trait condition (usually 60% HgbA). In addition, over half (5/9 mice) achieved sustained hHgbA of >70% at 20 weeks post-transplant. This is important because 70% HgbA is commonly used as a

clinical threshold for ameliorating SCD symptomology in transfusion protocols[29]. However, total Hgb levels in the PB remained lower than controls (Table 1), suggesting that the high INDEL frequencies may be reducing total Hgb production.

We evaluated other hematological parameters in reconstituted recipients at 16-weeks post-transplantation (Table 1). Of interest to our studies, the degree of myeloid donor chimerism in the allo-HSCT setting has been postulated as a surrogate for alleviating SCD symptoms[27,28]. We therefore investigated correlations between *HBB* gene correction and RBC functional parameters. There was a clear selective advantage in RBC development when comparing the frequency of myeloid gene correction with hHgbA percentages in the blood, with as much as a 30-fold increase observed (Fig. 2F). Consistent with the closer developmental relationship between myeloid and erythroid lineages however, myeloid gene correction and PB reticulocyte frequencies (a marker of erythropoietic output) showed a strong inverse correlation with an $R^2 = 0.81$ (Fig. 2G). Positive correlations between myeloid gene correction and hHgbA percentage ($R^2 = 0.74$) and total HGB levels ($R^2 = 0.54$) were also observed (Supplementary Fig. 2F, G). In addition, inverse correlations were observed between the percentage of hHgbA and PB reticulocytes ($R^2 = 0.6397$), highlighting that higher correction within RBCs also reduced erythropoietic stress (Supplementary Fig. 2H). By contrast, while often used as a surrogate for human HSPC engraftment in xenograft transplantation assays, the frequency of B-cell gene correction poorly correlated with PB reticulocyte frequencies ($R^2 = 0.0582$) (Supplementary Fig. 2I).

To experimentally confirm that *HBB* correction was achieved in functionally LT-HSCs, we performed secondary transplantation assays. Allele correction could be confirmed by ddPCR in myeloid (nm), B cell, and T cell PB lineages (Fig. 2H). In addition, hHgbA could also be detected in these secondary recipients (Fig. 2H), indicating erythroid reconstitution potential of gene-corrected LT-HSCs. These results confirm that *HBB* correction can be achieved in functionally multipotent LT-HSCs. Notably, INDELs were also detected in secondary recipients, indicating that LT-HSCs also correct HiFi Cas9-mediated DNA damage via non-homologous end joining (NHEJ) (Supplementary Fig. 2J).

### *HBB* correction prevents premature RBC turnover and sickling. We performed a detailed evaluation of in vivo RBC turnover and sickling following autologous HSCT of *HBB*-corrected HSCs, with a focus on a cohort of three recipient mice where 2 mice (mouse A and B) displayed high frequencies of myeloid gene correction and stable hHgbA, while the third (mouse C) did not (Table 1). Consistent with the selective advantage of HgbA-expressing RBCs in SCD mice[30–32], we observed ~10× longer

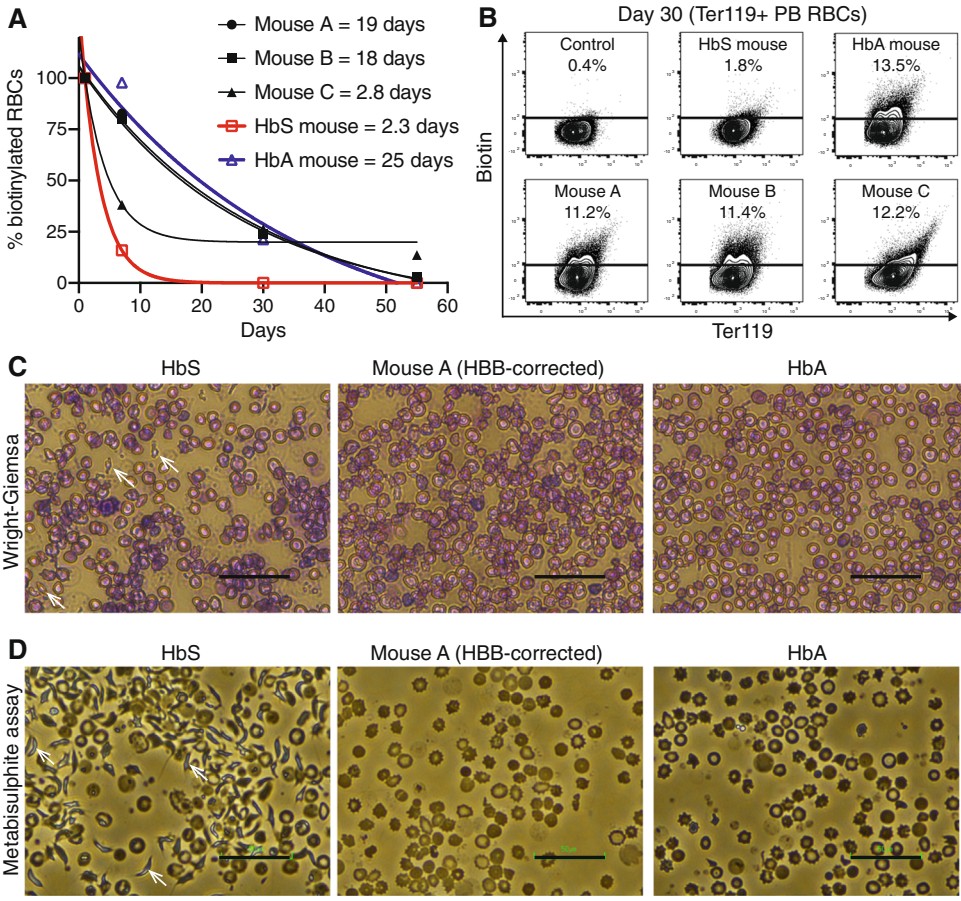

**Fig. 3 Autologous HSCT using E6V-corrected SCD HSCs can correct erythropoiesis in vivo. A** Half-life of EZH-biotin labeled Ter119+ PB RBCs, relative to labeling 24 h post-injection of EZH-biotin (RBC labeling efficiency at ~50%). Non-transplanted *HbS* and *HbA* mice included as controls. **B** Representative flow cytometric plots for Biotin vs Ter119 for Ter119+ PB RBCs at day 30 post-injection of EZH-biotin displayed in (**A**) and a no EZH-biotin injection control. **C** Representative Wright-Giemsa staining for PB from a non-transplanted *HbS* mouse, gene-corrected HSCT recipient (Mouse A), and a non-transplanted *HbA* mouse. Examples of abnormal RBC morphology indicated with white arrows. Representative for PB from Mouse A and Mouse B. Black scale bars indicate 50 μm. **D** Representative images of metabisulphite sickling assays performed on PB from a non-transplanted *HbS* mouse, gene-corrected HSCT recipient (Mouse A), and a non-transplanted *HbA* mouse. Abnormal RBC morphology indicated with white arrows. Representative for PB from Mouse A and Mouse B. Black scale bars indicate 50 μm. Source data are available in the Source data file.

RBC half-lives in gene-corrected HSCT recipients (Fig. 3A, B). Since a major cause of morbidity and mortality in SCD is caused by the sickling of RBCs, we evaluated RBC morphology using Wright-Giemsa staining and sodium metabisulphite sickling assays (Fig. 3C, D). In recipients with the high levels of HgbA, we further confirmed reduced frequency of abnormal RBC morphology (Fig. 3C) and reduced RBC sickling (Fig. 3D). Based on these data, we therefore conclude that autologous transplantation of Cas9-AAV6 *HBB*-corrected HSCs can substantially reduce the RBC abnormalities associated with SCD in vivo.

## Discussion

CRISPR/Cas9-based HSC gene correction is currently being developed as a potentially curative therapy for a range of congenital blood diseases including SCD[3,6,7], severe combined immunodeficiency diseases[9,33], metabolic diseases[34], and others[4]. However, evaluation of the functional consequences of these gene correction approaches on in vivo hematopoiesis is often lacking. To date, most studies have been performed using human HSPCs and rely on xenograft assays, which bias lineage reconstitution toward the B-cell lineage and poorly reconstitute RBC and platelet lineages (the two most abundant and essential PB components)[35]. Although mouse models are being developed to better support human hematopoiesis, such as human cytokine-

expressing mice[36,37] or the creation of a human niche[38,39], these models still do not fully recapitulate human hematopoiesis. The power of mouse models for pre-clinical validation was recently highlighted by Schiroli et al., who established a humanized SCID-X1 mouse and used it to evaluate a Cas9-based gene correction therapy[9]. Our results also highlight the potential pre-clinical benefit of using humanized mouse models for studying hematological diseases in vivo and evaluating the safety, feasibility, and efficacy of human HSC gene targeting reagents currently being developed as investigational cell products. Here, we demonstrate that serially repopulating mouse HSCs (with B, T, myeloid, PLT, and RBC output) can be gene targeted at both the *Rosa26* safe harbor locus and the humanized beta-globin gene locus using HiFi Cas9-AAV6 technology. We believe that this system will provide a useful tractable model to identify key parameters for eliciting high HR frequencies in functional LT-HSCs that will complement human HSPC studies. Our data also suggest areas for further improvement. In particular, the heterogeneity in engraftment chimerism points toward oligoclonality in gene editing of functional HSCs. Interestingly, this has also been recently observed in human HSPC gene editing via barcoding analysis[20,21], suggesting this is likely to become an important area for improvement as these approaches move toward the clinic.

Unlike current human HSC culture conditions, we can now stably maintain mouse HSCs ex vivo over several weeks[16,17]. We made use of this approach to grow mouse HSCs ex vivo for 7-days prior to gene editing, which boosted HR frequencies at the *Rosa26* locus by ~2.5-fold (Fig. 1C). Although this protocol differs from current human HSC gene editing protocols, given that HR frequencies increased with longer ex vivo expansion, we hope that improvements in human HSC culture conditions[40,41] will soon allow equivalent ex vivo culture times to be evaluated in the human setting. One limitation of the current study is how this extended culture might alter the function (and oligo-clonality) of the gene-corrected HSCs and will be a subject of studies that go beyond the scope of this current work.

Another advantage of performing gene targeting experiments in mouse HSCs is the ability to perform autologous HSCT, in order to assess gene correction frequencies in all of the blood and immune system lineages and evaluate the ability of gene-edited HSCs to support long-term recipient survival. However, few studies have taken advantage of established humanized mouse models to evaluate how autologous gene-edited HSCs engraft and function in the context of a diseased recipient. Here, by using autologous transplantation in combination with transfusion, we could assess the therapeutic benefit of Cas9-AAV6 corrected HSCs in SCD recipients. In this context, we could demonstrate that Cas9-AAV6 correction of the human E6V SCD-causing *HbS* gene mutation reduced erythropoietic stress as well as RBC sickling and turnover (hallmarks of SCD). It will be interesting to use this autologous transplantation model in the future to directly compare the efficacy of the various SCD gene therapy currently being developed. In addition, it will be important to study how the wider pathology of SCD including organ damage etc., is altered by *HBB*-gene correction and other autologous transplantation-based gene therapies.

One major finding of this study was the positive correlation between gene correction frequencies in the myeloid lineage and stable HgbA expression, and with phenotypic correction of in vivo erythropoietic indices, including PB reticulocyte frequencies and RBC turnover. While the HSC input correction frequencies were only ~20%, stable and similar levels of myeloid chimerism were maintained long-term in most of the transplanted animals. On average, we saw a ~9-fold selective advantage in the RBC compartment, due to the longer half-life of corrected RBCs. This could be an early insight into the correction frequencies needed to improve SCD patients in a clinical setting. For example, in the setting of allo-HSCT, donor cell chimerism that is >20% in the myeloid lineage show improved clinical symptoms with long-term stable HgbA expression[27,28,42]. It will be interesting to see if this translates to the setting of auto-HSCT using gene-corrected HSPCs.

One consideration when performing Cas9-based gene correction is the potential for loss-of-function causing INDELs within the targeted gene, caused by non-HR-based gene repair (e.g., NHEJ). Our data suggest that INDELs within *HBB* are retained long-term in vivo and likely contributed to the observed reductions in total hemoglobin levels in the transplanted mice. Although the retention of these mutant alleles did not cause a transfusion-dependent beta-thalassemia-like disease (where the mice need transfusions to survive)[43], it will be important to monitor this in the clinical setting. These data highlight the potentially deleterious side-effects of HSC HR-based gene correction strategies, and one that we should aim to minimize through further protocol optimization, e.g., incorporation of recently described NHEJ inhibitors[44]. One important point to consider is that the frequencies of gene targeting in this manuscript are ~3-times lower than what has been reported in human HSPCs[3,6,10,11,33,34], which means that the input of INDEL alleles

is substantially higher (i.e., a HR:INDEL ratio of 1:3 rather than 3:1). Nonetheless, it will be important to monitor such parameters during clinical development. In the future, it will also be interesting to investigate approaches to reduce the formation of these loss-of-function mutations during the gene-editing process[44–46] and limit the toxicity of gene editing on HSCs[10].

In summary, our study demonstrates that functional multipotent LT-HSCs can be gene-edited using Cas9-AAV6 technology and that beta-globin gene correction in SCD HSCs improves in vivo RBC indices following autologous transplantation. Together, our data support the clinical investigation of Cas9-AAV6 gene correction therapies while also identifying areas for future optimization of this powerful technology.

## Methods

**Mice**. All animal experiments were approved by the Administrative Panel on Laboratory Animal Care at Stanford University. C57BL/6-CD45.1 (PepboyJ; 002014), C57BL/6-CD45.2 (000664), and Townes-SCD (013071) mice were purchased from Jackson Laboratories. C57BL/6-CD45.1/CD45.2 F1 mice were bred in house. All mice were housed at ambient temperature with a 12-h light/dark cycle in a specific pathogen-free (SPF) condition with free access to food and water. Donor and recipient mice were used at 8–12 weeks old.

**HSC isolation**. CD150$^+$CD34$^-$Kit$^+$Sca1$^+$Lineage$^-$ HSCs were isolated from mouse BM by fluorescence-activated cell sorting (FACS)[16,17]. Briefly, WBMCs were stained with APC-cKit antibody and cKit-positive cells enriched using anti-APC magnetic beads and LS columns (Miltenyi Biotec). The cKit-enriched cells were then stained with a biotin lineage antibody cocktail (biotin-CD4, -CD8, -CD45R/B220, -TER119, -Gr1, and -CD127), before stained with anti-CD34, anti-cKit, anti-Sca1, anti-CD150, and streptavidin-APC/eFluor780 for 90 min (see Supplementary Table 1 for antibody details). Cells were purified using a FACS AriaII (BD) and BD FACS Diva 8 software by direct sorting into wells containing HSC media using PI as a live/dead stain (see Supplementary Fig. 3A for representative FACS gating).

**HSC culture**. HSC were cultured in media composed of F12 media (Life Technologies), 1% insulin-transferrin-selenium-ethanolamine (ITSX; Life Technologies), 1% penicillin/streptomycin/glutamine (P/S/G; Life Technologies), 10 mM HEPES (Life Technologies), 0.1% polyvinyl alcohol (Sigma P8136), 100 ng/ml thrombopoietin, and 10 ng/ml stem cell factor on fibronectin-coated plates (Corning)[16,17]. On day 6 after sorting, media was replaced with fresh pre-warmed media. On day 7, cells were collected for gene editing, as detailed below. After 12–18 h, media was replaced to remove gene editing reagents, and media was replaced every 2 days thereafter. HSC cultures were analyzed for GFP expression on day 14 by flow cytometry using a BD FACS AriaII or BD Fortessa. Flow cytometry data were analyzed using FlowJo 10 software and statistical analysis performed using Prism 7. Alternatively, cKit-enriched bone marrow was cultured in the same media for 48 h before gene editing. Media changes were performed after 12–18 h and cell cultures were then analyzed on day 4 by flow cytometry.

**Cas9-AAV6 gene editing**. Gene editing was performed by electroporation using a Lonza Nucleofector (pulse code EO100) and Solution P3 (Lonza) according to the manufacturer's protocol. Briefly, recombinant HiFi Cas9 (IDT) and synthetic sgRNAs (IDT or Synthego; see Supplementary Table 2) were pre-complexed at room temperature for 15 min and then diluted in P3 Solution (for 100 μl P3 Solution, 30 μg Cas9 and 16 μg sgRNA was used). Cells were collected, counted, and pelleted at 400 × $g$, and then resuspended in the P3 Solution containing Cas9/sgRNA and electroporated. Cells were returned to fresh pre-warmed HSC media and AAV6-repair template (generated by Vigene) added at a concentration of 5000 vector genomes/cell. The *HBB*-targeting AAV6 vector has been published previously[3]. *Rosa26*-targeting AAV6 was generated by cloning a Ubc promoter driven Zeon-Green expression cassette between *Rosa26* homology arms using primers described in Supplementary Table 3 and inserted into the AAV6 backbone[3]. For clonal analysis, gene-edited cells were plated into Methocult assays (M3434; STEMCELL Technologies) following manufacturer instructions and colonies picked after 6 days for ddPCR analysis.

**C57BL/6-congenic HSC transplantation**. *Rosa26*-targeted HSCs from C57BL/6-CD45.1 mice were injected intravenously (IV) alongside 2 × 10$^5$ or 1 × 10$^6$ WBMCs from C57BL/6-CD45.1/CD45.1 mice into female C57BL/6-CD45.2 mice following lethal-dose irradiation (9.5 Gy). Donor chimerism was tracked by collecting peripheral blood (PB) cells and staining with anti-CD45.1, anti-CD45.2, anti-CD11b, anti-Ly-6G/Ly-6C, anti-CD45R (B220), anti-CD4, anti-CD8 antibodies for 30 min (see Supplementary Table 1 for antibody details). Following a wash step, cells were analyzed by flow cytometry and/or FACS. PI was used as a live/dead stain for leukocyte analysis. See Supplementary Fig. 3B and Fig. 1D for representative FACS

gating. Separately, PB cells were stained with anti-CD41, anti-CD42, and anti-Ter119 for 30 min (see Supplementary Table 1 for antibody details) and analyzed by flow cytometry. Secondary BM transplantation assay was performed by transferring $1 \times 10^6$ BM cells from the primary recipient mice into lethally-irradiated C57BL/6-CD45.2 mice. Donor chimerism was analyzed as above.

**Townes SCD autologous HSC transplantation**. Gene-corrected HSCs from female donor Townes-SCD mice were transplanted into female recipient Townes-SCD mice following lethal-dose irradiation (9.5 Gy). Recipient mice received IV injections of ~50 μl of mouse peripheral blood PB prior to radiation and on 3, 7, and 10 days after radiation. Donor chimerism was tracked by collecting PB cells as described above, and sorting CD11b/Ly-6G$^+$ myeloid cells, CD4/CD8$^+$ T cells, and CD45R$^+$ B cells for ddPCR, as described in Supplementary Fig. 3B. Secondary BM transplantation assay was performed by transferring $1 \times 10^6$ BM cells from the primary recipient mice into lethally-irradiated C57BL/6-CD45.1 mice. Donor chimerism was analyzed as above.

**Droplet digital PCR**. Droplet digital PCR (ddPCR) was used to quantify genomic editing frequencies using genomic DNA extracted using QuickExtract DNA Extraction Solution (Epicenter). To quantify *Rosa26* gene editing, ddPCR reactions for the GFP insert were set up using the *Rosa26-GFP* in-out and genomic reference primer/probe sets described in Supplementary Table 4, with the genomic reference primer/probe set used to quantify GFP detection. Droplets were generated and analyzed according to manufacturer's instructions using the QX200 system (Bio-Rad). ddPCR cycling conditions were as follows: 95 °C (10 min); followed by 50 cycles of 94 °C (30 s), 60 °C (1 min), and 72 °C (2 min); and then 98 °C (10 min). Data were analyzed using Biorad Quantasoft 1.7.

Quantification *HBB* gene editing was performed using the primer/probe sets described in Supplementary Table 4 by ddPCR[26]. Briefly, genomic DNA was extracted using QuickExtract DNA Extraction Solution and PCR amplicons (length of 1410 bp) spanning the targeted region were generated using the *HBB* in-out primer pair. PCR cycling conditions for in-out PCR are as follows: 98 °C (30 s); followed by 35 cycles of 98 °C (10 s), 60 °C (30 s), and 72 °C (1 min); and finally 72 °C (10 min). The PCR products were run on a 1% agarose gel and the band located at 1410 bp was cut and purified using QIAquick gel extraction kit (Qiagen). Subsequent PCR product was diluted to 10 ng/μl. Subsequently, 6 serial dilutions of the PCR products were made in nuclease-free $H_2O$ down to 10–20 fg/μl, which served as the template DNA for the ddPCR reaction. ddPCR reactions were set up using a *HBB* ddPCR primer pair and HR-(HEX), REF-(HEX), and WT-(FAM) probes. Droplets were generated and analyzed according to manufacturer's instructions using the QX200 system (Bio-Rad). ddPCR cycling conditions were as follows: 98 °C (10 min); followed by 50 cycles of 94 °C (30 s), 60 °C (30 s), and 72 °C (2 min); and then 98 °C (10 min).

**Blood parameters**. Complete blood counting was performed using a Horiba Micros ESV 60. Hemoglobin tetramers were quantified by high-performance liquid chromatography (HPLC). Reticulocyte assays were performed using Retic-Count reagent (BD) according to the manufacturer's protocol. RBC sickling assays were performed by mixing 2 μl PB with 2 μl of freshly prepared 2% sodium metabisulphite solution (Sigma) on a glass slide, applying a coverslip, sealing, and incubating for 2–4 h. RBC half-life studies were performed by injecting 50 mg/kg of NHS-biotin IV in recipient mice (>16-weeks post-transplantation) and tracking the percentage of biotin$^+$ RBCs in the PB by flow cytometry following PE-streptavidin and APC/eFluor780-Ter119 antibody staining.

**Reporting summary**. Further information on research design is available in the Nature Research Reporting Summary linked to this article.

## Data availability
Data supporting the findings of this work are available within the paper and its Supplementary Information files. A reporting summary for this Article is available as a Supplementary Information file. The datasets and materials generated and analyzed during the current study are available from the corresponding author upon request. Source data are provided with this paper.

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

## Acknowledgements
We thank the Stanford Stem Cell Institute FACS Core for flow cytometry access, the Stanford BioADD lab for performing HPLC assays, and F. Suchy for advice on ddPCR assays. H.N. was supported by CIRM (LA1_C12-06917) and the NIH (R01DK116944; R01HL147124). A.C.W. was supported by the NIH (K99HL150218) and the Leukemia and Lymphoma Society (3385-19). D.P.D. was supported by the NIH (R01HL135607). M.H.P. gratefully acknowledges the support of the Amon Carter Foundation, the Laurie Kraus Lacob Faculty Scholar Award in Pediatric Translational Research and the NIH (R01AI097320; R01AI120766).

## Author contributions
A.C.W. and D.P.D. conceptualized the research, performed and supervised experiments, analyzed data, and wrote the manuscript. R.B., J.C., I.H., C.C., and C.M. performed experiments, analyzed data, and edited the manuscript. M.H.P. and H.N. conceptualized and directed the research, participated in the design and interpretation of the experiments, and wrote the manuscript.

## Competing interests
H.N. is a co-founder and shareholder in ReproCELL, Megakaryon, and Century Therapeutics. M.H.P. has equity and serves on the scientific advisory board of CRISPR Therapeutics and Allogene Therapeutics. However, none of these companies had input into the design, execution, interpretation, or publication of the work in this manuscript. All other authors do not declare any competing interests.
