## [Peer Review File · Nature Communications]

REVIEWERS' COMMENTS

Reviewer #2 (Remarks to the Author):

Overall, the Authors have suitably and satisfactorily revised the manuscript and tempered their original claims to better fit the actual findings and their experimental context. Also, the discussion has been substantially improved and now better highlights both the relevance and the limitations of the study.

It is unfortunate that the authors did not supplement the manuscript with some assessment of bone marrow histopathology and hematopoietic lineage composition, which would help understanding to what extent the improved hematological parameters observed in the peripheral blood are reflected in the hematopoietic organs. This Reviewer remains convinced that such data would be helpful to better assess the benefit of edited cells administration also in view of the counteracting effect of globin gene disruption by NHEJ in a fraction of cells, the overall replication stress imposed on the progenitors and the observed anemia phenotype. I would however leave the final decision whether such additional data should be incorporated in the final version of the manuscript to the Editor sensitivity about the current challenges faced by the authors in setting up new *in vivo* studies.

There are also few minor points that should be easily addressed

The Authors now cite the “Schiroli et al., 2017” paper as asked by another Reviewer but they are not correct when describing it, as in this work the Authors have “humanized” the ILR2G gene and thus used editing reagents targeting the human sequence.

When describing the oligoclonal reconstitution by human edited HSPC they should also cite, in addition to their BioRxiv manuscript, a paper recently published by the Naldini’s lab (Ferrari et al, Nat Biotech 2020) reporting similar findings.

Signed by: Luigi Naldini

Point-by-point response to Reviewer comments on NCOMMS-20-40078-T

Reviewer #2 (Remarks to the Author):

Overall, the Authors have suitably and satisfactorily revised the manuscript and tempered their original claims to better fit the actual findings and their experimental context. Also, the discussion has been substantially improved and now better highlights both the relevance and the limitations of the study. It is unfortunate that the authors did not supplement the manuscript with some assessment of bone marrow histopathology and hematopoietic lineage composition, which would help understanding to what extent the improved hematological parameters observed in the peripheral blood are reflected in the hematopoietic organs. This Reviewer remains convinced that such data would be helpful to better assess the benefit of edited cells administration also in view of the counteracting effect of globin gene disruption by NHEJ in a fraction of cells, the overall replication stress imposed on the progenitors and the observed anemia phenotype. I would however leave the final decision whether such additional data should be incorporated in the final version of the manuscript to the Editor sensitivity about the current challenges faced by the authors in setting up new in vivo studies.

We would like to thank Reviewer 2 for re-reviewing our manuscript and their thoughtful comments on our work. We have corrected the errors noted below, as described.

There are also few minor points that should be easily addressed:

The Authors now cite the “Schiroli et al., 2017” paper as asked by another Reviewer but they are not correct when describing it, as in this work the Authors have “humanized” the ILR2G gene and thus used editing reagents targeting the human sequence.

We apologize for this error and have removed the erroneous sentence from the Discussion.

When describing the oligoclonal reconstitution by human edited HSPC they should also cite, in addition to their BioRxiv manuscript, a paper recently published by the Naldini’s lab (Ferrari et al, Nat Biotech 2020) reporting similar findings.

We apologize for missing this key reference and have now added it to the manuscript (new reference 21).